# SEE, THINK, HALLUCINATE: INTERPRETING REASONING AND HALLUCINATIONS BEYOND THE FIRST HOP IN VISION-LANGUAGE MODELS

## ABSTRACT

Vision-language models (VLMs) are prone to hallucinations, including errors such as factual inaccuracies, biases, and reasoning failures. Prior research has primarily focused on object hallucinations in single-hop settings, where models are asked to describe an image and are evaluated on whether they mention non-existent objects. However, such work overlooks broader forms of hallucination that arise in more complex reasoning scenarios. In this paper, we investigate hallucinations in vision-language reasoning *beyond the first hop*, where models must first extract factual content from an image and then combine it with external knowledge to answer a question. In particular, we first present `MMHop`, a dataset of multimodal two-hop questions spanning five knowledge categories: general reasoning, perceptual co-occurrence, temporal knowledge, cultural and regional knowledge, and biasd prior knowledge. Using `MMHop`, we conduct a systematic analysis of VLMs with different architectures and LLM backbones, uncovering where hallucinations arise and how reasoning unfolds. Our comparative study reveals distinct failure tendencies: some models are easily distracted by visual co-occurrence, while others rely excessively on internal knowledge or stereotypical priors. Beyond model-specific behaviors, our results highlight common structural patterns in two-hop reasoning. VLMs exhibit a two-stage inference process: an *input understanding stage*, dominated by multi-head attention, followed by a *reasoning stage*, where feed-forward networks become increasingly important. Early reasoning layers primarily capture first-hop inference, while later layers focus on second-hop reasoning. We further identify failure modes across categories: shortcut reliance on visual context, shallow recall of temporal knowledge, weak cultural grounding, and bias-driven errors. Finally, we show that question variants and inference settings, such as test-time scaling, can alter reasoning dynamics and reduce hallucination. Our analyses provide new interpretability-driven insights into multimodal hallucinations, paving the way toward more reliable and trustworthy vision-language reasoning systems.

## 1 INTRODUCTION

Vision–Language Models (VLMs) (Achiam et al., 2023; Liu et al., 2023; Team et al., 2024; Meta, 2024) have achieved remarkable progress across a wide range of multimodal tasks, including image captioning, visual question answering, and complex multimodal reasoning. Despite these advances, VLMs remain prone to *hallucinations*, generating outputs that are fluent yet factually incorrect, ungrounded in the image, or inconsistent with external knowledge (Zou et al., 2024; Li et al., 2025). Such errors undermine trustworthiness in everyday use and pose serious risks in high-stakes domains such as medical imaging or autonomous driving, where reliable multimodal reasoning is critical.

Prior research has primarily focused on hallucinations in single-hop settings, where models are asked to describe an image or answer a straightforward question, and are evaluated on whether they mention non-existent or incorrect objects (Huo et al., 2024; Cho et al., 2025; Zhang et al., 2025). While important, this captures only a narrow subset of hallucinations. Real-world reasoning often extends *beyond the first hop*, requiring models to extract intermediate facts from an image and then integrate them with external or commonsense knowledge, as illustrated in Figure 1. In such settings, hallucinations are not limited to object-level errors but also arise from deeper reasoning

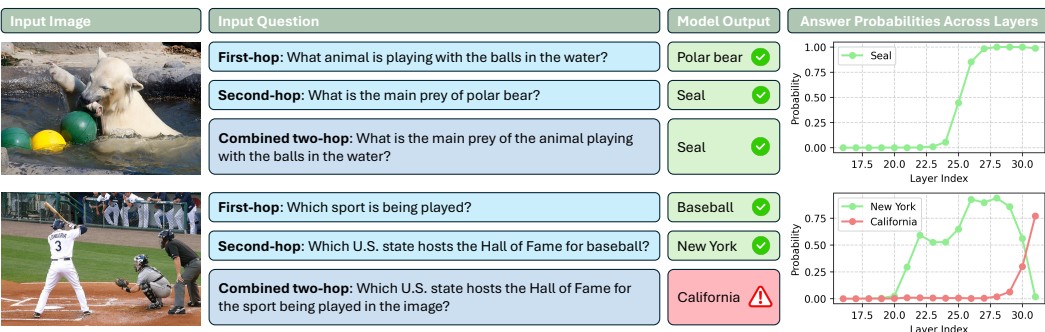

Figure 1: Correct and incorrect multimodal two-hop reasoning. In the first example, the model answers both hops correctly and successfully integrates object identification with external knowledge, yielding the correct combined answer. The decoded answer probabilities rise steadily for the correct token across layers. In the second example, although the model answers each hop correctly in isolation, it fails on the combined question, showing a breakdown in reasoning integration. The decoded probabilities initially favor the correct answer but are overtaken by the incorrect one in the final layers.

failures: misattributing visually co-occurring cues, overlooking temporal changes, ignoring cultural specificity, or defaulting to stereotypical priors. Understanding these richer forms of hallucination is essential for building more reliable multimodal systems.

In this paper, we move beyond single-hop evaluation and investigate hallucinations in *multimodal two-hop reasoning*. To enable systematic study, we introduce `MMHop`, a new dataset of two-hop multimodal questions spanning five knowledge categories: general reasoning, perceptual co-occurrence, temporal knowledge, cultural and regional knowledge, and biasd prior knowledge. Through our preliminary study, we reveal two key insights. First, VLMs follow a two-stage inference process: an *input understanding stage* and a *reasoning stage*. Second, within the reasoning stage, earlier layers primarily resolve first-hop inference, while deeper layers focus on second-hop reasoning. Building on these findings, we design new metrics to characterize hallucinations across categories, uncovering failure modes such as over-reliance on visual cues, shortcut reasoning without intermediate steps, weak cultural grounding, and bias-driven errors.

**Contributions.** Our work makes the following contributions: (1) We introduce `MMHop`, a dataset of multimodal two-hop questions designed to elicit and analyze hallucinations across five distinct knowledge categories. (2) Through systematic interpretability experiments, we reveal the internal reasoning dynamics of VLMs, showing a two-step inference process where early layers ground first-hop reasoning and deeper layers consolidate second-hop reasoning. (3) We propose novel metrics and provide a detailed error analysis of VLM hallucinations, revealing category-specific failure modes and offering new insights into the internal mechanisms behind multimodal hallucinations. (4) We conduct additional experiments, including test-time scaling and hidden states probing, uncovering valuable findings that further illuminate the dynamics of multimodal reasoning and hallucination. Together, these contributions advance both benchmarking and interpretability of VLMs, laying the foundation for more reliable multimodal reasoning systems.

## 2 PRELIMINARIES OF VLM INFERENCE AND GENERATION

**Input representation.** VLMs process multimodal inputs by combining visual and textual information in a unified transformer architecture. Formally, the input consists of an image $I$ and a text prompt $X = \{x_1, \ldots, x_m\}$. The image $I$ is first encoded by a vision encoder into a sequence of visual embeddings $\{v_1, \ldots, v_p\}$, which are then projected into the text embedding space through a projection layer. The resulting visual tokens $\{v_i\}$ are concatenated with the text tokens $\{x_j\}$, forming the joint input sequence $z^0 = \{v_1, \ldots, v_p, x_1, \ldots, x_m\}$, which is fed into the LM.

**Transformer inference.** The joint sequence $z^0$ is processed by a stack of $N$ transformer decoder layers. Each layer $\ell \in \{1, \ldots, N\}$ consists of a multi-head self-attention (MHA) module and a feed-forward network (FFN), both applied with residual connections:

$$\tilde{z}^\ell = \text{MHA}^\ell(z^{\ell-1}) + z^{\ell-1}, \quad z^\ell = \text{FFN}^\ell(\tilde{z}^\ell) + \tilde{z}^\ell.$$

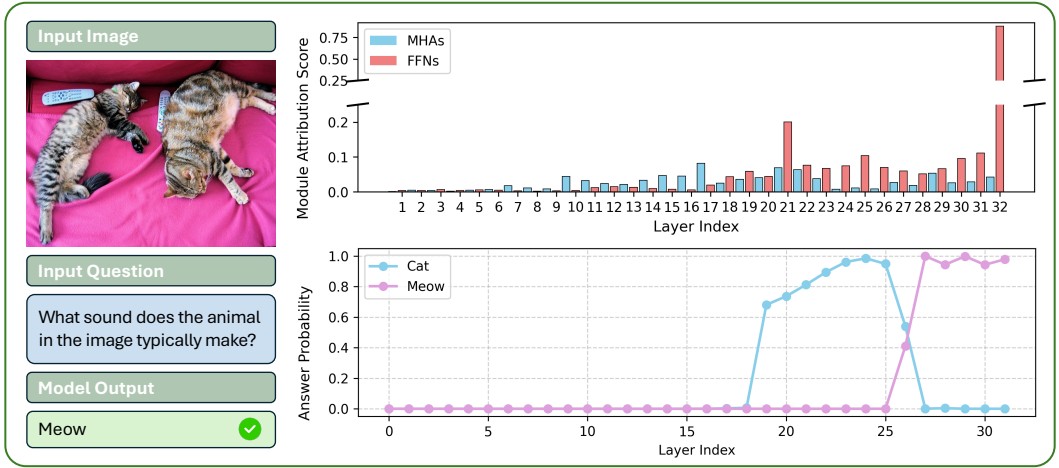

Figure 2: VLM reasoning dynamics. The example illustrates a two-hop question where the model must first identify the object "cat," retrieve knowledge about its sound, and integrate the two steps to answer correctly. The top-right figure shows module attribution scores: early layers are dominated by MHAs, indicating integration of visual and textual input, while later layers are dominated by FFNs, reflecting reliance on parametric knowledge. The bottom-right figure shows token probabilities across layers: first-hop answers ("cat") emerge earlier, while second-hop answers ("meow") rise in later layers, demonstrating step-by-step two-hop reasoning.

The residual stream $z^\ell$ therefore accumulates contributions from both MHAs and FFNs. Intuitively, MHAs aggregate information across tokens, including cross-modal interactions, while FFNs inject and apply parametric knowledge stored in the network's weights through nonlinear transformations.

## 3  HOW DO VLMS PERFORM TWO-HOP REASONING?

In this section, we conduct a preliminary study to investigate how VLMs perform two-hop reasoning. To this end, we manually construct 100 two-hop questions from images sampled from COCO (Lin et al., 2014) and analyze VLM behavior using attribution metrics and interpretability tools.

### 3.1  METRICS DESIGN

**Module contribution attribution.** Since each module writes additively into the residual stream, we quantify its influence by measuring the extent to which it shifts hidden states. Specifically, we compute attribution scores using the Wasserstein distance between the residual representations before and after each module:

$$\text{Attr}^\ell_{\text{MHA}} = W(z^{\ell-1}, \tilde{z}^\ell), \quad \text{Attr}^\ell_{\text{FFN}} = W(\tilde{z}^\ell, z^\ell),$$

where $W(\cdot, \cdot)$ denotes the Wasserstein distance. Larger distances correspond to stronger shifts in representation space, and thus greater influence of the module on the model's predictions.

**Logit lens.** To probe intermediate representations across layers, we adopt the *logit lens* technique. Given a hidden state $z^\ell_t$ at layer $\ell$ for position $t$, the logit lens projects it into the vocabulary distribution using the unembedding matrix $W_U$:

$$p_\ell(y \mid z^\ell_t) = \text{softmax}(\text{LayerNorm}(z^\ell_t)W_U).$$

Although only the final-layer hidden states are trained to predict the next token, applying the logit lens to intermediate states reveals prediction tendencies across layers, providing a lens into the model's evolving reasoning. In our experiments, we focus on the position of the first answer token and omit the index $t$ for simplicity.

### 3.2  FINDINGS

**VLMs exhibit a two-stage inference process of input understanding and reasoning.** Figure 2 (top right) shows attribution scores across layers for LLaVA. A consistent depth-wise pattern emerges. In the early layers, multi-head attention (MHA) dominates residual stream updates, indicating that the model is primarily integrating multimodal inputs, including both visual and textual

tokens. In later layers, feed-forward networks (FFNs) become the primary contributors, suggesting increasing reliance on parametric knowledge to reason about the question and refine predictions. Attention remains active throughout, but the clear crossover from MHA-heavy to FFN-heavy updates marks a transition from input contextualization to knowledge-driven reasoning. This two-stage process is robust across architectures, backbones, and prompt variations in our two-hop setting.

**Early reasoning layers capture first-hop inference, while later layers focus on second-hop reasoning.** We further probe reasoning dynamics by decoding intermediate hidden states with the logit lens. Figure 2 (bottom right) shows token probabilities for a commonsense two-hop question: first identifying the animal ("cat") and then recalling its characteristic sound ("meow"). We find that token probabilities start to appear in the second reasoning stage identified above, and they reveal a sequential progression of reasoning. In earlier reasoning layers, the model converges on the first-hop answer, with "cat" rapidly increasing in probability and dominating alternatives. In deeper layers, the probability of "cat" declines as "meow" rises and eventually dominates, indicating a shift from entity recognition to factual recall. This depth-wise progression shows that two-hop reasoning is distributed across layers: early ones resolves the first hop, while later layers handle the second hop.

**Summary.** Together, these findings reveal a structured inference process in VLMs: (i) an *input-understanding stage*, where MHAs dominate and contextualize multimodal inputs, followed by (ii) a *reasoning stage*, where FFNs drive major representational shifts and finalize predictions. Within the reasoning stage, logit-lens trajectories further decomposes the process into first-hop inference in earlier layers and second-hop reasoning in later layers. This layered structure provides a mechanistic explanation of how VLMs solve two-hop questions and lays the foundation for our subsequent analysis of hallucination patterns across knowledge categories.

# 4 METHODOLOGY

Building on the insights from Section 3.2, we aim to systematically investigate how VLMs perform two-hop reasoning and where hallucinations arise. To this end, we design a set of inspection metrics that quantify different aspects of the reasoning trajectory. By combining these perspectives, we obtain a fine-grained view of when intermediate answers appear, how strongly they persist, and whether the reasoning chain collapses. This allows us not only to localize hallucinations to specific stages of inference but also to distinguish between different failure modes across question categories.

## 4.1 METRICS FOR REASONING AND HALLUCINATION ANALYSIS

Building on the preliminary study in Section 3.2, we design five inspection metrics to quantify how VLMs perform two-hop reasoning and where hallucinations emerge. These metrics are derived from the logit lens, which decodes intermediate hidden states across layers into vocabulary distributions, allowing us to trace when and how candidate answers appear during inference. Together, they capture complementary aspects of reasoning quality, failure modes, and hallucination dynamics.

**Context Reliance Score (CRS).** Attention heads primarily retrieve contextual information from image and text tokens, while FFNs apply parametric knowledge encoded in weights. To assess whether hallucinations arise from over-reliance on visual context, we measure whether the generated answer first emerges through an MHA or an FFN update.

For each example $i$, let $y_i^{\text{gen}}$ denote the generated answer token. We define an indicator $M(y_i^{\text{gen}})$, which checks whether the *first position in the residual stream $r$* at which the probability of $y_i^{\text{gen}}$ exceeds a threshold $\alpha$ arises from an MHA update rather than an FFN update:

$$M(y_i^{\text{gen}}) = \begin{cases} 1, & \text{if the first } r \text{ with } p_i^r(y_i^{\text{gen}}) > \alpha \text{ occurs after MHA,} \\ 0, & \text{if it occurs after FFN.} \end{cases}$$

Let $\mathcal{H}$ and $\mathcal{C}$ denote hallucinated and correct cases of sizes $N_H$ and $N_C$, respectively. We compute

$$M_H = \frac{1}{N_H} \sum_{i \in \mathcal{H}} M(y_i^{\text{gen}}), \quad M_C = \frac{1}{N_C} \sum_{i \in \mathcal{C}} M(y_i^{\text{gen}}),$$

and define CRS $= M_H - M_C$. A higher CRS indicates hallucinations are more likely to emerge from attention-based context integration, suggesting that the model over-relies on visual cues rather than grounding its reasoning in parametric knowledge stored in FFNs.

**First-Step Activation Score (FAS).** FAS evaluates whether a model internally activates the first-hop answer even if the final output is a hallucination. For hallucinated instance $i$, let $y_i^{\text{1st}}$ denote the first-hop answer. We record whether its probability ever exceeds a confidence threshold $\beta$:

$$\text{FAS} = \frac{1}{N_H} \sum_{i=1}^{N_H} \mathbf{1} \left[ \max_\ell \ p_i^\ell(y_i^{\text{1st}}) > \beta \right].$$

A higher FAS suggests that the model partially follows the reasoning chain before diverging.

**Reasoning Collapse Score (RCS).** RCS measures whether a model fails to differentiate between the two reasoning steps of a two-hop question. If the same token is generated for both the first-hop and second-hop answers, it indicates that the model bypasses the intended reasoning chain and simply repeats the same concept twice. Formally:

$$\text{RCS} = \frac{1}{N_H} \sum_{i=1}^{N_H} \mathbf{1} \left( y_i^{\text{1st}} = y_i^{\text{2nd}} \right).$$

A high RCS suggests that the model struggles with two-hop reasoning, producing trivial or circular answers instead of applying second-hop knowledge.

**First Commitment Score (FCS).** FCS captures how early the final generated token $y_i^{\text{gen}}$ first becomes active. For each hallucinated case $i$, let $L_i$ be the first layer where $p_i^\ell(y_i^{\text{gen}}) > \alpha$. With $N$ total layers,

$$\text{FCS} = \frac{1}{N_H} \sum_{i=1}^{N_H} \left( 1 - \frac{L_i}{N} \right).$$

A higher FCS indicates early commitment, while a lower FCS indicates delayed emergence.

**Answer Persistence Score (APS).** APS measures how long the generated answer remains dominant once it emerges. For each hallucinated case $i$, let $D_i$ be the number of layers where $p_i^\ell(y_i^{\text{gen}}) > \beta$:

$$\text{APS} = \frac{1}{N_H} \sum_{i=1}^{N_H} \frac{D_i}{N}.$$

High APS indicates persistent dominance of the final answer across layers, while low APS reflects unstable reasoning. Together, FCS and APS capture complementary aspects of commitment: *when* an answer first appears and *how long* it persists.

## 5 MMHOP: A DATASET FOR MULTIMODAL TWO-HOP QUESTIONS

To systematically study hallucinations in two-hop multimodal reasoning, we construct MMHop, a benchmark designed to probe how VLMs integrate visual grounding with external knowledge. Unlike prior datasets that primarily emphasize single-hop object recognition or caption-based evaluation, MMHop focuses on compositional reasoning across two steps: (i) extracting factual information from an image, and (ii) combining it with external commonsense or factual knowledge to answer a question. By systematically covering diverse knowledge categories and using carefully curated question designs, MMHop provides a controlled yet challenging testbed for analyzing how VLMs handle multi-step inference and where hallucinations arise. In this section, we describe our two-stage automatic data generation pipeline, the knowledge categories we consider, and the dataset statistics that demonstrate its scalability and diversity.

**First-hop data generation.** We construct the first hop using images from the COCO 2017 validation split. For each image, we begin with the provided captions and use an LLM to select a specific object that is clearly mentioned. To enhance diversity and difficulty, the LLM is encouraged to choose objects that are easily confusable with plausible but absent alternatives (e.g., "motorcycle" vs. "bicycle"), so that distinguishing them requires precise grounding. The object must also have easily differentiable attributes or facts that can be leveraged in the second hop. The resulting first-hop questions focus on object identification, requiring the model to ground its reasoning in the image.

**Knowledge categorizations and second-hop data generation.** Once the first-hop object is identified, we generate second-hop questions that require reasoning over knowledge not visible in the

Figure 3: Data generation pipeline for `MMHop`. From COCO images and captions, an LLM selects an object and forms a first-hop question. Then we instruct the LLM to integrate external knowledge to build two-hop questions across 5 categories, producing diverse question–answer pairs grounded in both vision and knowledge.

image. Each two-hop question is aligned with one of the following five knowledge categories, designed to capture distinct reasoning requirements and potential failure modes. (1) *General Two-Hop Knowledge* serves as an umbrella category, where the second hop requires commonsense or factual knowledge beyond the image. (2) *Perceptual Co-Occurrence Knowledge* focuses on distinguishing between what co-occurs visually in the image and what is factually true, preventing models from relying on shortcuts based on co-occurring visual context. (3) *Temporal Knowledge* requires reasoning over historical changes in objects, technologies, or practices, testing whether the model can differentiate past from present contexts. (4) *Cultural and Regional Knowledge* highlights questions whose answers differ across cultural or geographic contexts, testing whether models can avoid defaulting to globally averaged responses. (5) *Bias and Prior Knowledge* targets questions where stereotypical or overgeneralized associations can mislead the model, requiring it to override spurious priors with contextually accurate reasoning.

For each first-hop object, we use an LLM to retrieve facts aligned with these categories and to construct corresponding two-hop questions. To avoid trivial formulations, the second-hop question must not be directly answerable from the image alone, and it must clearly separate the correct object from its plausible alternatives. The LLM is also instructed to generate complete sets of acceptable answers: for instance, numerical answers must include both word-form and Arabic-form variants (e.g., "two" and "2").

This pipeline balances scalability with quality control: questions are automatically generated but constrained by prompts enforcing diversity, difficulty, and category coverage. The resulting dataset spans a wide range of multimodal reasoning questions, grounded in images but requiring reasoning beyond perception. Each instance includes an image, a first-hop question with possible answers, a second-hop question with possible answers, and a combined two-hop question. Dataset statistics are provided in Table 3 in Section A.

## 6 EXPERIMENTS

**Experimental setup.** We evaluate `MMHop` across 8 open-weight VLMs that span different architectures, backbone LLMs, series versions, and parameter scales. We also cover 2 API-based models: Claude 3.5 Haiku and GPT-4.1 mini. For each model, we independently ask the first-hop question, the second-hop question, and the combined two-hop question. To probe the internal reasoning process, we instruct the model to directly output the answer without any verbal illustrations. To explicitly assess knowledge availability, the second-hop question is asked *without* providing the image, ensuring that the answer depends solely on the model's internal or parametric knowledge.

For each instance in `MMHop`, we consider a model correct only if it answers all three questions accurately, measured by exact match. We define this as *Full-Chain Accuracy (FCA)*, which captures the end-to-end success rate across perception, knowledge retrieval, and reasoning.

For hallucination analysis, we focus on two specific error modes: (1) *Knowledge Insufficiency Rate (KIR)* — the proportion of cases where the model correctly answers the first-hop question but fails on both the second-hop and the combined two-hop question. This indicates that the model can ground

the object in the image but lacks the external knowledge needed for the second step. (2) **Reasoning Failure Rate (RFR)** — the proportion of cases where the model correctly answers both the first-hop and second-hop questions in isolation but fails to answer the combined two-hop question. This reveals that, while the model has access to the relevant perceptual and factual knowledge, it struggles to integrate them into a coherent reasoning chain.

**Evaluation results.** The performance of all models on `MMHop` is reported in Table 1. Since the dataset is constructed with GPT-5, API-based models achieve relatively high scores and are shown primarily for reference. Among the open-weight models, however, FCA remains modest: even the strongest model reaches only about 60%, meaning that failures occur on nearly half of the examples for most models. Larger models generally perform better, reflecting the benefits of scale, but the improvements remain insufficient to overcome the challenges of multi-step reasoning beyond the first hop.

For hallucinated cases, both KIR and RFR appear consistently in the 10–20% range. KIR reflects cases where models correctly identify objects in the image but lack the external knowledge needed for the sec-

Table 1: Performance of different models on `MMHop`. FCA: Full-Chain Accuracy, KIR: Knowledge Insufficiency Rate, RFR: Reasoning Failure Rate, ↑/↓: higher/lower the better.

| Model | FCA ↑ | KIR ↓ | RFR ↓ |
|---|---|---|---|
| Llava-1.5-7B | 53.12 | 16.29 | 18.03 |
| Llava-1.6-Vicuna-7B | 56.00 | 16.73 | 15.35 |
| Llava-1.6-Mistral-7B | 58.96 | 16.17 | 13.71 |
| Llava-1.6-Llama3-8B | 60.74 | 15.15 | 12.73 |
| Qwen2-VL-2B | 5.52 | 28.71 | 10.72 |
| Qwen2-VL-7B | 53.98 | 18.57 | 15.03 |
| Qwen2.5-VL-3B | 45.32 | 23.05 | 17.11 |
| Qwen2.5-VL-7B | 41.16 | 22.51 | 10.30 |
| Claude 3.5 Haiku | 70.01 | 5.15 | 11.04 |
| GPT-4.1 mini | 76.74 | 4.36 | 5.06 |

ond hop. RFR is particularly striking: even when models succeed on both hops individually, they frequently fail on the combined two-hop question, underscoring difficulties in integrating perceptual grounding with factual knowledge into a coherent reasoning chain. Taken together, these results demonstrate that hallucinations are not isolated to specific architectures or backbones, but instead represent a systematic limitation shared across model families and sizes.

## 7 WHY DO VLMS HALLUCINATE?

In Section 3, we examined how VLMs process different stages of reasoning beyond the first hop. Building on these insights, we now take a step further to investigate the *hallucination patterns* that emerge within VLMs. Using `MMHop` as a benchmark, we inspect model internal states with the techniques introduced in Section 4. For hallucination analysis, we focus exclusively on the *reasoning failure* cases defined in Section 6, where the model correctly answers both the first-hop and second-hop questions in isolation but fails on the combined two-hop question. This isolates hallucinations that arise not from perceptual grounding errors or missing knowledge, but from failures to integrate available information into a coherent reasoning chain. All reported results are averaged across the 8 open-weight VLMs evaluated in the previous section.

**VLMs are prone to visual distractions and shortcut reliance.** Across general and perceptual co-occurrence questions, models show high CRS. This indicates that VLMs are easily distracted by visual cues that are correlated but factually irrelevant, often defaulting to shortcut strategies such as selecting co-occurring objects instead of reasoning through the two-hop chain. These failures reveal that models overweight visual grounding at the expense of logical integration, leading to answers that appear plausible from the image alone but are incorrect when external knowledge is required.

**VLMs struggle to carry out step-by-step reasoning in temporal knowledge questions.** Temporal questions expose a different weakness: models frequently bypass intermediate reasoning, as reflected in low FAS. Instead of explicitly reasoning through the first-hop object and then applying temporal knowledge, models often attempt to recall an answer directly from internal parametric memory. This pattern suggests that VLMs lack robust mechanisms for chaining temporal information, defaulting to shallow recall rather than structured reasoning.

**VLMs show weak grounding in cultural and regional reasoning.** When cultural or region-specific knowledge is required, VLMs often fail to incorporate it into their reasoning chain. While they may successfully resolve the first hop, as indicated by high FAS, their high RCS reveals that they frequently repeat the first-hop answer, ignoring the required cultural or regional specificity. This reveals a systematic weakness in grounding second-hop reasoning beyond generic or globally averaged priors.

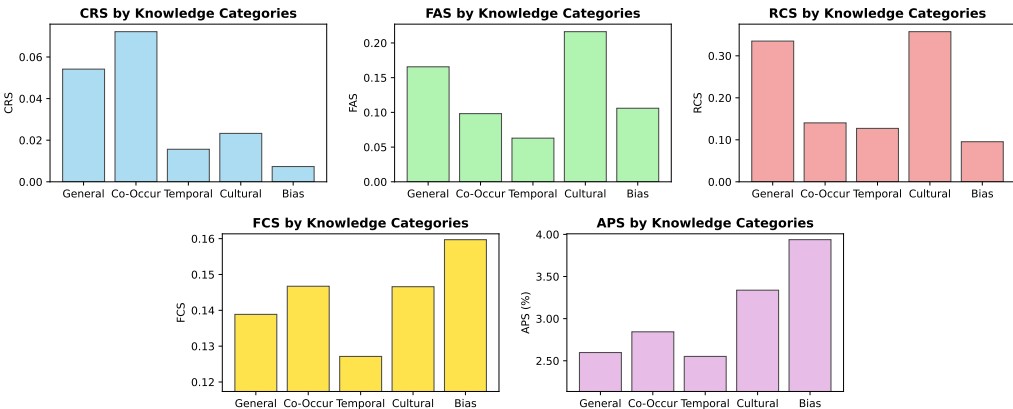

Figure 4: Hallucination analysis across five metrics. Each plot reports results averaged over 8 open-weight VLMs on `MMHop`. CRS (top left) shows reliance on visual context, FAS (top middle) captures whether first-hop answers are activated, RCS (top right) measures collapse into repeated answers, FCS (bottom left) indicates how early models commit to a final answer, and APS (bottom right) quantifies how persistently answers dominate across layers. Together, these metrics reveal distinct failure modes across knowledge categories.

**VLMs rely heavily on internal biases and stereotypical priors.** Bias-driven questions present another severe failure mode. Models rarely consult visual grounding signals, as indicated by low CRS, but instead commit early to biased answers, reflected in high FCS. Once these answers emerge, they dominate across layers, leading to the highest APS. This shows that hallucinations driven by stereotypes and priors are not only early but also persistent, with little chance of correction during reasoning. Such reliance on stereotypes illustrates a mode of hallucination where reasoning is replaced by memorized patterns.

**Summary.** Together, these findings illustrate that hallucinations in multimodal reasoning are not uniform, but arise from diverse failure modes: shortcut reliance on visual context, shallow recall in temporal reasoning, weak incorporation of cultural knowledge, and persistent bias-driven errors. By disentangling these categories, our analysis provides a finer-grained understanding of why VLMs hallucinate and where progress is most urgently needed.

## 8 ADDITIONAL FINDINGS

**Test-time scaling slightly mitigates reasoning failures.** In Section 6, we restricted models to produce direct answers, measuring their internal reasoning capacity. To assess the effect of test-time scaling (TTS), we now allow models to generate free-form responses. Table 2 reports Full-Chain Accuracy (FCA) with and without TTS. Across most models, FCA improves by 3–6 points, showing that TTS helps models articulate intermediate reasoning steps and reach more accurate two-hop answers. However, gains remain modest and uneven: while some Qwen variants even degrade under TTS, larger open-weight models like LLaVA consistently benefit. This suggests that TTS can alleviate, but not eliminate, systematic reasoning failures in multimodal two-hop tasks.

Table 2: Performance of different models on `MMHop`. FCA: Full-Chain Accuracy, TTS: Test-time scaling.

| Model | FCA + TTS | Δ FCA |
|---|---|---|
| Llava-1.5-7B | 57.06 | 3.94 |
| Llava-1.6-Vicuna-7B | 61.38 | 5.38 |
| Llava-1.6-Mistral-7B | 63.81 | 4.86 |
| Llava-1.6-Llama3-8B | 65.21 | 4.48 |
| Qwen2-VL-2B | 2.78 | -2.74 |
| Qwen2-VL-7B | 58.58 | 4.60 |
| Qwen2.5-VL-3B | 53.10 | 7.78 |
| Qwen2.5-VL-7B | 35.83 | -5.34 |
| Claude 3.5 Haiku | 73.41 | 3.40 |
| GPT-4.1 mini | 78.60 | 1.85 |

**Linear probing reveals that VLMs encode question complexity.** Linear probing provides a simple yet powerful lens into the internal representations of VLMs. Using `MMHop`, we investigate whether models internally distinguish between single-hop and two-hop questions. We split the dataset into training and test sets and train linear probes on hidden states from different layers of Llava-1.5-7B. The results, shown in Figure 5, demonstrate that probing accuracy rapidly improves in early layers and remains near-perfect across later layers.

This indicates that VLMs encode the complexity of input questions very early in their processing pipeline, and that this information is robustly preserved throughout the network.

## 9 RELATED WORK

**Hallucinations in Vision–Language Models.** Hallucinations are prevalent in VLMs, spanning visually ungrounded mentions, attribute errors, and reasoning failures where outputs are fluent yet unsupported by visual evidence or external facts (Huang et al., 2025). Most prior work concentrates on object hallucinations in single-hop settings, such as captioning or simple VQA (Li et al., 2023; Chen et al., 2024). Recent work has begun to connect such errors to specific internal mechanisms. Ferrando et al. (2024) use sparse autoencoders to reveal directions encoding knowledge awareness in LMs, showing causal effects on refusal or hallucination. Yang et al. (2025) perform modular ablations to attribute hallucination propensity to particular components. DAMO (Wang et al., 2025b) and DeCo (Wang et al., 2025a) trace layer-wise probability trajectories and show that models may internally recognize correct objects earlier, then drift in deeper layers. Our

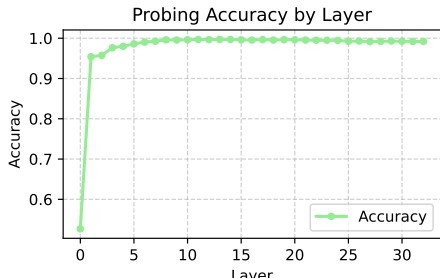

Figure 5: Probing accuracy in Llava-1.5-7B. A linear probe is trained on hidden states from each layer to classify whether the input is a single-hop or two-hop question. Accuracy rises sharply in the early layers and remains high across depth, showing that question complexity is encoded early and preserved throughout the network.

work complements these efforts by shifting the focus *beyond the first hop*: rather than only diagnosing object presence errors, we analyze multimodal reasoning hallucinations that arise when models must integrate grounded visual facts with external knowledge, and we relate these errors to module-level dynamics through layerwise attribution and logit-lens probing.

**Multi-hop Reasoning in Vision–Language Models.** Multi-hop reasoning in text-only LLMs has been extensively explored through datasets requiring intermediate facts (Yang et al., 2018) and prompting strategies such as chain-of-thought (Wei et al., 2022). For VLMs, however, benchmarks isolating multi-hop composition and hallucination remain limited. GQA (Hudson & Manning, 2019) and A-OKVQA (Schwenk et al., 2022) emphasize compositional questions over attributes and spatial relations, but primarily require image-grounded chains with limited external knowledge. We-bQA (Chang et al., 2022) integrates images with retrieved web snippets, yet answers are typically contained in the retrieved context, making external knowledge explicit in the input rather than latent in the model. By contrast, our benchmark targets reasoning *beyond single hop*: the first hop extracts an image-grounded fact, while the second hop requires external or commonsense knowledge absent from the image, spanning perceptual co-occurrence, temporal, cultural/regional, and prior-knowledge dimensions. This design enables controlled diagnosis of where hallucinations originate within the reasoning chain and how they map to internal computations, thereby complementing prior datasets with explicit links between error types and interpretable model behavior.

## 10 CONCLUSION

In this work, we presented `MMHop`, a new benchmark for studying hallucinations in multimodal reasoning beyond single-hop object recognition. By spanning five categories of external knowledge and analyzing eight diverse VLMs, we revealed systematic patterns in how models process two-hop questions, including a two-stage inference process with distinct roles for attention and feed-forward modules and category-specific failure modes such as shortcut reliance, weak temporal reasoning, poor cultural grounding, and bias-driven errors. Our interpretability-driven approach sheds light on the internal dynamics of hallucinations and offers a principled framework for diagnosing multimodal reasoning failures, though it is limited by its reliance on the non-zero logit lens, which may not generalize to specific model architectures or decoding schemes. We hope that `MMHop` and our findings inspire future work on both more robust benchmarks and new mechanistic methods that broaden the scope of hallucination analysis, ultimately paving the way toward more trustworthy and reliable vision-language models.

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

Table 3: Statistics of the MMHop dataset. We report the number of examples in each category.

| Category | General | Co-Occur | Temporal | Cultural | Bias | Total |
|---|---|---|---|---|---|---|
| **Size** | 2212 | 1184 | 737 | 493 | 376 | 5002 |

## A  MMHOP STATISTICS AND IMPLEMENTATION DETAILS

We show the statistics of MMHop in Table 3. The LLM in Figure 3 we use to generate the dataset is GPT-5. We collect $5,002$ two hop questions across 5 different knowledge categories. We also provide the prompt we use to generate data below.

---

**Prompts for data generation**

```
You will be given an image and a list of captions describing the
image.  Your task is to generate two-hop questions based on the
provided image and captions.  These questions will be used to study
hallucinations in vision-language models.

Guidelines for generating the questions:
- Each two-hop question consists of two sub-questions ("hops"):
1.  First-Hop Question:  This should be a question about a specific
object in the image and mentioned in the captions.  Identify
the object and rephrase the combined information into a concise
question, avoiding excessive modifiers.  The selected object
should have characteristics similar to other common objects not
mentioned in the captions.  It's preferable to choose objects and
their potential incorrect alternatives that have unique, easily
distinguishable differences or different facts.
2.  Second-Hop Question:  Ask a question about factual knowledge
related to the identified object that cannot be directly answered
from the image or captions.  This question should differentiate the
correct object from its incorrect alternatives, requiring external
or common knowledge.

For the given image, generate one two-hop question per category
listed below.  If the image doesn't contain relevant content for
a particular category, skip it.  Prioritize generating fewer,
high-quality questions rather than attempting all categories at
the expense of clarity.  Note that for most cases, only the first
category is applicable.

{CATEGORY_DEFINITION}

Format your responses as JSON objects (one JSON object per line)
with these keys:

- question_type:  (int) Category number of the question type.
- first_hop_question:  The first-hop question about the identified
object.
- first_hop_answer:  List of correct names or synonyms of the
selected object.
- first_hop_wrong_answers:  List of plausible but incorrect
alternative objects similar to the correct answer.
- second_hop_question_template:  A template for the second-hop
question that involves factual knowledge about the object without
information from the captions.  The template should include a
placeholder for the object name.
- full_question:  The full question that combines the first-hop
question and the second-hop question template.
```

---

**Prompts for data generation (cont.)**

```
- second_hop_answer:  The correct answer to the second-hop question
based on the identified object.
- second_hop_wrong_answers:  A list of incorrect answers to the
second-hop question based on the incorrect alternatives.

When generating questions and answers, it's preferable to generate
answers with only one word.  For the "first_hop_answer" and
"second_hop_answer", provide possible names of the selected object
and potential correct names.  For the incorrect alternatives,
provide a list of objects similar to the correct answer.

Now, provide your response in the same JSONL format using the
following image and captions.  Only include the JSON objects (one
per line) in your response, ensuring each answer includes synonyms
or related keywords where applicable.
{IMAGE}
Captions:  {CAPTION}
```

