# OpenReview forum: "See, Think, Hallucinate: Interpreting Reasoning and Hallucinations Beyond the First Hop in Vision-Language Models"
_ICLR.cc/2026/Conference — Submitted to ICLR 2026_

### Official Review · Reviewer_pn9j · 2025-10-28

**Soundness:** 2
**Presentation:** 2
**Contribution:** 2
**Rating:** 2
**Confidence:** 4

**Summary:**

This paper introduces MMHop, a new dataset for evaluating hallucinations in multimodal two-hop reasoning, moving beyond existing single-hop benchmarks.
Authors find that VLMs use a two-stage inference process, where early layers resolve the first reasoning hop and deeper layers handle the second.
Based on these findings, the paper proposes new metrics and provides a detailed error analysis, identifying specific failure modes like shortcut reasoning and over-reliance on visual cues.

**Strengths:**

The narrative is compelling: the authors first conduct a preliminary study (Section 3) to discover a two-stage inference process and the layer-wise separation of reasoning hops. They then build upon these findings to design their novel interpretability metrics (Section 4) and the MMHop benchmark (Section 5).

The primary novelty of this work is its conceptual shift in VLM hallucination research. It moves the problem from "perception failure" (seeing non-existent objects) to "reasoning failure" (failing to integrate correct perception with external knowledge).

**Weaknesses:**

1. The paper provides a strong analysis of where hallucinations occur but stops short of proposing solutions. Given the deep interpretability findings (e.g., identifying specific FFN/MHA layers as sources of failure), a significant weakness is the lack of a proposed mitigation method. The paper would be more impactful if it leveraged these insights to suggest or test a solution, such as a targeted fine-tuning strategy or an inference-time intervention.

2. The analysis in Figure 4 is averaged across all 8 open-source VLMs. This approach may obscure crucial differences between model architectures (e.g., LLaVA vs. Qwen), which might have fundamentally different internal reasoning processes. Averaging these results could lead to conclusions that don't hold true for individual models.

3. The claims regarding test-time scaling (TTS) in Table 2 seem weakly supported. The impact on FCA is highly inconsistent, with results ranging from a 7.78-point gain to a 5.34-point loss. This variance makes it difficult to draw a firm conclusion about the effectiveness of this method.

4. The exploration of prompting strategies is limited. The paper does not explore stronger methods, such as explicit Chain-of-Thought (CoT) prompting, which would be a natural fit for a multi-hop reasoning task.

5. The experimental scope regarding model scale seems insufficient. The analysis of scaling laws appears limited to 3B and 7B models. To make a more convincing claim, it would be beneficial to include experiments on larger models, such as the Qwen-VL 32B or even larger variants like 235B.

6. The authors state that the dataset is constructed with GPT-5 but there is no mention of a human validation step. This makes it difficult to assess the quality, correctness, and potential biases of the dataset. Furthermore, the dataset statistics provided are minimal; crucial details like the distribution of first-hop objects or the difficulty distribution of second-hop knowledge are missing.

7. Typo: There appears to be a typo ("biasd" --> "biased") in the abstract and in the introduction (line 77).

**Questions:**

The interpretability finding of a two-stage reasoning process is a key contribution. Is there evidence to suggest this holds true for other VLM architectures beyond the LLaVA and Qwen families tested in this paper?

Regarding the dataset construction (Table 3), the sample distribution across categories is quite unbalanced. The general category has the most samples, which is easy to understand, but why is there such a large difference between the other categories? Could the authors provide a rationale for this distribution?

In Figure 3, the "motorcycle" example is very helpful, but it is not specified which of the five knowledge categories the final generated question belongs to. Could the authors clarify this and perhaps show the full generation process for this sample?

---

### Official Review · Reviewer_MPe3 · 2025-10-29

**Soundness:** 2
**Presentation:** 3
**Contribution:** 3
**Rating:** 4
**Confidence:** 4

**Summary:**

This paper studies two-hop reasoning and hallucination in VLMs. It introduces MMHop, a dataset of two-hop multimodal questions spanning five categories, generated from COCO images via an LLM-based pipeline. Using layerwise attribution and logit lens decoding, the authors report a consistent two-stage pattern: early MHAs dominate input understanding, while later FFNs dominate knowledge-driven reasoning; first-hop answers peak earlier, and second-hop answers later.

**Strengths:**

- **Comprehensive experiment analysis.** The experiments are clearly described, with well-defined metrics and thorough explanations. The authors evaluate a broad set of models (eight in total, both open- and closed-source) under consistent conditions, and the dataset spans five diverse topics, ensuring fair and comprehensive analysis.
- **Dataset contribution.** The introduced dataset could be helpful for the community.
- **Clear and well-presented paper.** The writing is clear, well structured, and easy to follow, with strong visualizations and coherent explanations that effectively communicate the main findings.

**Weaknesses:**

- **Heavy reliance on the logit lens at a single position.** The analysis focuses on the first token of the final answer, which risks position-specific artifacts and may overlook multi-token dynamics or earlier positions where information is consolidated (especially for subword tokenization or multi-word answers). The paper acknowledges focusing on “the position of the first answer token” for simplicity, but this may overstate the generality of the “two regions at the last token” effects. CRS asks whether the first emergence of the generated token crosses a threshold after an MHA vs. FFN update. When computed only at the last token position, it is vulnerable to the possibility that earlier-token MLP content is transported by attention in later layers (as prior work shows for “moving” features via attention in LLMs [1]), which can misattribute the underlying source. The metric, as used, does not disambiguate write-site vs. move-site.
- **Insufficient dataset quality control.** Given the synthetic nature of the dataset, it would be beneficial to include some form of quality control or validation. For example, this could be done using a different LLM as a judge or, preferably, a small-scale human evaluation to assess accuracy and category fidelity.
- **Contradiction between probing and reasoning claims.** The linear probe achieves near-perfect accuracy, distinguishing question complexity by layers 5-10 (Figure 5), yet the paper claims reasoning primarily occurs in later layers via FFNs (Section 3.2). This temporal mismatch raises questions about whether the probe detects causally relevant semantic complexity or merely superficial linguistic features, and the paper provides no evidence (e.g., through ablation) that early complexity encoding actually influences downstream reasoning.

- **Missing causal analysis.** The paper claims distinct functional roles for attention and MLP layers, but does not test these claims in a causal way. It would strengthen the work to evaluate whether predictions change when selectively removing or perturbing these components. *However, this point is just a minor point and does not affect my judgment.*


[1] Geva et. al., Dissecting Recall of Factual Associations in Auto-Regressive Language Models, 2023

**Questions:**

- **Reproducibility of the dataset.** Please specify which version of GPT-5 was used, the associated generation cost, and key sampling parameters, as this would enhance reproducibility.


- **Choice of threshold.** Clarify the rationale for selecting these threshold values (alpha and beta) and, if possible, discuss the sensitivity of results to their variation.

---

### Official Review · Reviewer_q5yQ · 2025-10-31

**Soundness:** 3
**Presentation:** 3
**Contribution:** 3
**Rating:** 6
**Confidence:** 4

**Summary:**

The authors propose MMHop, a new benchmark for two-hop multimodal reasoning, where models must first extract information from an image (first hop) and then combine it with external knowledge (second hop). The paper identifies that VLMs exhibit a two-stage inference process. Through systematic experiments and interpretability analyses (e.g., logit lens, attribution scores), the paper uncovers where and why hallucinations emerge. It introduces several diagnostic metrics—Context Reliance Score, First-Step Activation Score, Reasoning Collapse Score, First Commitment Score, and Answer Persistence Score—to characterize different hallucination types. Results across eight open-weight VLMs and two API-based models show that even strong models struggle with multi-step reasoning, often failing to integrate visual grounding with factual knowledge. The authors further find distinct hallucination modes such as shortcut reliance on visual context, shallow temporal recall, weak cultural grounding, and bias-driven reasoning.

**Strengths:**

1. The paper extends hallucination analysis from single-hop perception tasks to multi-hop reasoning, addressing a largely unexplored yet practically important aspect of vision-language models (VLMs).
2. The proposed dataset systematically covers five reasoning categories (general, co-occurrence, temporal, cultural, and bias-related), enabling fine-grained evaluation of multimodal reasoning.
3. The use of layer-wise attribution and logit lens analysis provides valuable mechanistic insights into how attention and feed-forward modules contribute differently to perception and reasoning.
4. The proposed quantitative metrics (CRS, FAS, RCS, FCS, APS) offer a principled way to diagnose hallucination emergence and persistence across reasoning stages.

**Weaknesses:**

1. MMHop is generated via GPT-based prompting, which may inherit linguistic or cultural biases and limit the dataset’s authenticity and ecological validity.
2. The study focuses only on two-hop reasoning, which may not generalize to more complex, multi-step or open-domain reasoning chains.
3. While the analyses are descriptive, the paper lacks causal experiments (e.g., ablation or activation editing) to confirm the proposed reasoning-stage separation.
4. The work provides a comprehensive evaluation and interpretability framework but stops short of offering concrete methods to alleviate hallucinations.  Do the authors have some suggestions?

**Questions:**

My questions are mentioned in the weakness.

---

### Official Review · Reviewer_Epn3 · 2025-11-01

**Soundness:** 3
**Presentation:** 3
**Contribution:** 3
**Rating:** 4
**Confidence:** 5

**Summary:**

The paper investigates hallucination in the multi-hop reasoning of VLMs. First, the paper analyzes 100 two-hop questions from COCO images, using Logit lens. The analysis shows that VLMs generate the first-hop answer before the second-hop reasoning. To understand the multi-hop reasoning, the paper introduces five metrics.
- **Context Reliance Score (CRS)** measures the degree of hallucination after MHA. A high CRS means that MHA drives hallucinated answers.
- **First-Step Activation Score (FAS)** measures whether VLMs correctly reason the first-hop answer despite the incorrect second-hop answer. A high FAS means the first-hop reasoning is correct.
- **Reasoning Collapse Score (RCS)** measures if VLMs fail to progress to the second-hop answer from the correct first-hop answer.
- **First Commitment Score (FCS)** measures the layer at which the final answer is first determined.
- **Answer Persistence Score (APS)** measures the number of layers where the final answer remains consistent. A high APS means that the VLM's decision remains across layers.
The paper curates a new dataset, MMHop, consisting of 5002 samples. With this dataset, the five metrics, and VLMs, the paper gives a comprehensive analysis of two-hop reasoning and hallucination.

**Strengths:**

**S1. Fine-grained Metrics.** The paper introduces five metrics to gain a deeper understanding of VLMs' behavior. These metrics quantify the behavior of VLMs, as discussed in Sections 7 and 8.

**S2. Interesting Phenomena.** The proposed metrics reveal several interesting observations. For example, VLMs exhibit limitations in second-hop reasoning due to high FAS and RCS.

**S3. Paper writing.** The writing is well-structured.

**Weaknesses:**

**W1. Analysis of Methods to Mitigate Hallucination.** The paper analyzes VLMs, including LLaVA, Qwen, Claude Haiku, and GPT Mini. Several methods have been proposed to reduce hallucinations, including Contrastive Decoding (CD) and Latent Steering. Latent Steering [Nullu CVPR'25, VTI ICLR'25] modifies the latent representation, allowing the proposed metrics to be applied to these methods. This analysis offers valuable insights into their relative strengths and limitations, contributing to a deeper understanding of hallucination mitigation in VLMs.

**W2. Applicability of Metrics.** Contrastive Decoding (CD) generally modifies the probability of the last token during decoding. For closed VLMs (e.g., Claude, GPT), the internal representations are not publicly accessible. The proposed metrics do not apply to these methods and VLMs for analysis.

**W3. Samples of the Dataset.** While the data (JSONL) is provided in the supplementary material, including a sample visualization in the main paper would enhance the understanding of the proposed dataset.

**W4. Dataset Quality Control.** In the final step in the dataset curation, was the dataset subject to manual review by humans or experts?

**Questions:**

**Q1. Refer.** As I understand, in Section 7, each paragraph explains the results of Figure 4. However, the paper omits the reference to the Figure, making it difficult to follow the writing. Including the guidance, such as "As shown in ...", would improve the readability.

**Q2. Discussion of the recent work.** Recently, a multi-hop reasoning dataset [ReasonVQA ICCV'25] has been proposed. Could the authors elaborate on how the proposed dataset differs from the previous studies?

---

### Meta-Review · Area_Chair_L5E7 · 2025-12-15

**Summary:**

The paper received mixed evaluations with scores of 6, 4, 4, and 2. While the reviewers appreciated the fine-grained metrics and the analysis of the two-stage inference process, significant concerns were consistently raised regarding the reliability of the unverified synthetic dataset, the limited methodological scope (e.g., single-token logit lens analysis), and the absence of any proposed mitigation strategies for the identified hallucinations. As the authors did not submit a rebuttal to address these critical issues regarding the validity and practical contribution of the work, the decision is to reject this paper.

**Reviewer Concerns:**

Addressed: Clarifications regarding specific metric definitions and the rationale for parameter thresholds (e.g., alpha/beta values).
Outstanding: The lack of human verification for the synthetic dataset (MMHop), the absence of causal experiments to validate the "two-stage" claim, and the failure to propose concrete methods to alleviate the analyzed hallucinations.

**Reviewer Scores:**

Reviewer Epn3: 4

Reviewer q5yQ: 5

Reviewer MPe3: 4

Reviewer pn9j: 2

---

### Decision · Program_Chairs · 2026-01-26

Reject